# Directional selectivity of afferent neurons in zebrafish neuromasts is regulated by Emx2 in presynaptic hair cells

Young Rae Ji[1], Sunita Warrier[2], Tao Jiang[1], Doris K Wu[1]*, Katie S Kindt[2]*

[1]Section on Sensory Cell Regeneration and Development, Laboratory of Molecular Biology, Bethesda, United States; [2]Section on Sensory Cell Development and Function, National Institute on Deafness and Other Communication Disorders, National Institutes of Health, Bethesda, United States

**Abstract** The orientation of hair bundles on top of sensory hair cells (HCs) in neuromasts of the lateral line system allows fish to detect direction of water flow. Each neuromast shows hair bundles arranged in two opposing directions and each afferent neuron innervates only HCs of the same orientation. Previously, we showed that this opposition is established by expression of Emx2 in half of the HCs, where it mediates hair bundle reversal (*Jiang et al., 2017*). Here, we show that Emx2 also regulates neuronal selection: afferent neurons innervate either Emx2-positive or negative HCs. In emx2 knockout and gain-of-function neuromasts, all HCs are unidirectional and the innervation patterns and physiological responses of the afferent neurons are dependent on the presence or absence of Emx2. Our results indicate that Emx2 mediates the directional selectivity of neuromasts by two distinct processes: regulating hair bundle orientation in HCs and selecting afferent neuronal targets.

DOI: https://doi.org/10.7554/eLife.35796.001

*For correspondence:
wud@nidcd.nih.gov (DKW);
katie.kindt@nih.gov (KSK)

**Competing interests:** The authors declare that no competing interests exist.

## Introduction

The survival of any animal is dependent on accurate processing of environmental inputs via its sensory organs. Some aquatic animals have a specialized sensory organ or lateral line system that enables them to sense water pressure and the direction of water flow. The lateral line system is important for many survival behaviors such as prey detection, mating and predator avoidance (*Coombs and Conley, 1997*; *Montgomery et al., 1997*; *Engelmann et al., 2000*). This sensory system is comprised of individual neuromasts distributed over the surface of the body (*Figure 1A*). Each neuromast consists of a cluster of sensory HCs that sense the direction of water flow bi-directionally along either the anterior-posterior (A-P) or dorsal-ventral (D-V) axis of the body (*Ghysen and Dambly-Chaudière, 2004*).

Two structural organizations within neuromasts are essential for detecting bi-directional water flow. First, specialized hair bundles that protrude from the top of the HCs are structured asymmetrically to enable directional sensitivity (*Figure 1B*; *Flock and Wersall, 1962*; *López-Schier et al., 2004*). Each hair bundle is comprised of a staircase of actin-filled microvilli called stereocilia, located next to the kinocilium (*Figure 1A*). Only deflections of the bundle towards its kinocilium open mechanotransduction channels on top of the stereocilia, which allow entry of positive ions and result in activation (depolarization) of the HC (*Figure 1B*; *Shotwell et al., 1981*). By contrast, deflections of the hair bundle in the opposite direction away from its kinocilium close mechanotransduction channels and result in inhibition (hyperpolarization). Thus, the two opposite orientations of the hair bundles within a neuromast provide its bi-directional sensitivity.

The second structural organization required to detect directional flow within neuromasts are the neurons of the lateral line ganglion. These neurons innervate neuromast HCs such that each neuron only innervates HCs that exhibit the same hair bundle orientation (*Figure 1B*; *Nagiel et al., 2008*; *Faucherre et al., 2009*). Although earlier results seem to be in conflict (*Nagiel et al., 2009*; *Faucherre et al., 2010*), more recent work has consistently concluded that lateral line neurons select their HC targets independent of HC activity, based on studies in mutants that lack HC mechano-transduction or neurotransmission (*Nagiel et al., 2009*; *Pujol-Martí et al., 2014*). These studies reduced the likelihood of an activity dependent model for innervation selection. However, the molecular mechanism(s) that determine how this directional selectivity is preserved at the level of neuronal innervation is not known.

Previous results have demonstrated that the orientation of hair bundles within neuromasts as well as vestibular maculae of the inner ear in both zebrafish and mice is controlled by the transcription factor, Emx2 (*Jiang et al., 2017*). In neuromasts, as each HC precursor divides to form two nascent HCs, Emx2 is only expressed in one of the two daughter cells. As a result, Emx2 is expressed in half of the HCs within a neuromast, and it functions to change the hair bundle orientation to approximately 180 degrees from the default state to generate a bi-directional hair bundle pattern (*Jiang et al., 2017*). Due to the role of Emx2 in establishing directional selectivity at the HC level, we asked whether Emx2 is also involved in dictating this selectivity at the neuronal level. For our analyses, we examined the innervation pattern of single afferent neurons in both *emx2* knockout (ko) and gain of *emx2* function (gof, *myo6b:emx2-mCherry*) zebrafish mutants. Our results indicate that the expression of Emx2 in the HCs also enables afferent neurons to select their proper HC targets and preserve the directional selectivity of the neuromast. However, neuronal selectivity is not dependent on hair bundle orientation. In *vangl2$^{m209}$* mutants, in which hair bundles are randomly oriented, afferent neurons still segregate according to Emx2-positive and -negative HCs.

## Results

### Directional selectivity of afferent neurons is correlated with Emx2 expression in HCs

In a wildtype neuromast, each afferent neuron only innervates HCs with apical hair bundles polarized in one of the two orientations (*Nagiel et al., 2008*; *Faucherre et al., 2009*; *Pujol-Martí et al., 2014*). Emx2 is also expressed in HCs of just one orientation (*Jiang et al., 2017*). Based on these results, we hypothesized that each afferent neuron should only innervate either Emx2 positive or negative HCs within a neuromast (*Figure 1A–B*). To test this hypothesis, we injected a *neuroD:tdTomato* plasmid into transgenic *myo6b:actb1-GFP* zebrafish embryos. This enabled us to visualize single afferent neurons labeled with tdTomato innervating a neuromast and the orientation of hair bundles labeled with Actb1-GFP. *Figure 1* illustrates two such examples of wildtype neuromasts in the primary posterior lateral line (*Figure 1C–J,K–R*). These neuromasts contain HCs oriented along the anterior-posterior axis of the body. As previously demonstrated, in these neuromasts, Emx2 immunoreactivities are only detected in mature HCs with hair bundles oriented towards the posterior direction (A > P, *Figure 1C,F–J,K,N–R*; *Jiang et al., 2017*). Emx2 is also expressed in immature HCs that can be identified by weak GFP signals and/or incomplete bundle polarity establishment. Only one of the immature HCs within a pair expresses Emx2 (*Figure 1C–R*, yellow arrow). Notably, we found that each single-labeled neuron either associated with all the Emx2-positive HCs in a neuromast (example, *Figure 1C–J*), or associated with all the Emx2-negative HCs in a neuromast (example, *Figure 1K–R*). Based on the association of single afferent neurons with HCs that were either Emx2-positive or-negative, we hypothesized that Emx2, which is required for establishing the hair bundle orientation (*Jiang et al., 2017*), may also be involved in regulating the pattern of neuronal innervation.

### Neuronal innervation patterns are altered in *emx2* mutants

To test the above hypothesis, we investigated the neuronal innervation pattern in neuromasts of loss and gain of *emx2* function zebrafish. We first generated wildtype and *emx2* mutant fish in an

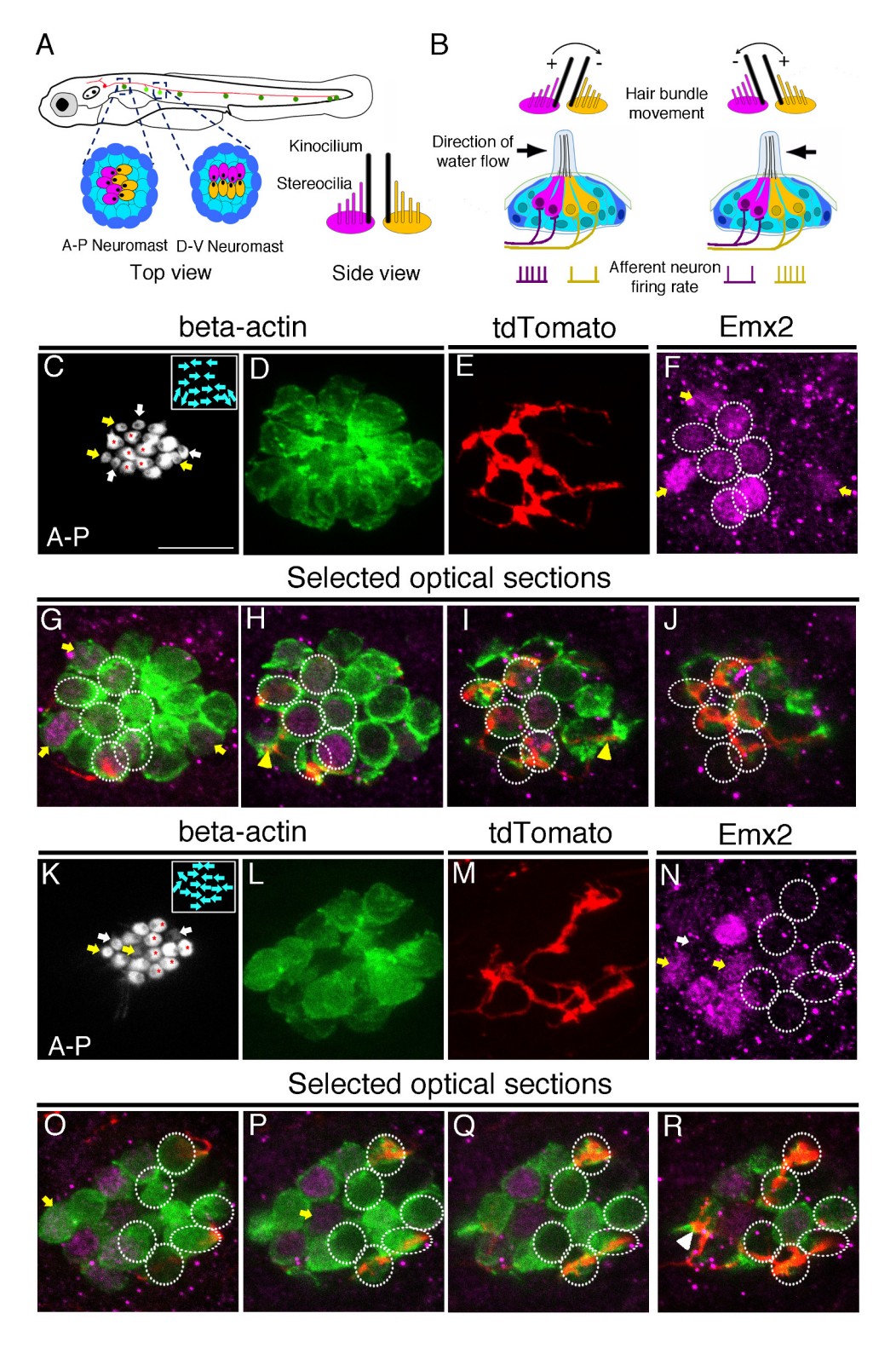

**Figure 1.** Afferent neuronal innervation is correlated with Emx2 expression and hair bundle orientation. (**A**) Schematic diagram of zebrafish neuromasts in the posterior lateral line showing HCs (magenta and yellow) in A-P (dark green) and D-V (light green) oriented neuromasts, which are innervated by neurons of the posterior lateral line ganglion (red). A side view of the hair bundles (kinocilium plus stereocilia) on the apical surfaces of HCs is shown. Only the posterior- or ventral-oriented HCs (magenta) are Emx2-positive. (**B**) Sensitivity to the direction of water flow within a neuromast is mediated by

*Figure 1 continued on next page*

*Figure 1 continued*

the two sets of HCs with opposite bundle orientations and their segregated neuronal innervations. (C–J) An afferent neuron (E) is associated with A > P (asterisks in (C)), Emx2-positive HCs ((F-J), circles; (H,I), yellow arrowhead) in a *myo6b:actb1-GFP* neuromast (C,D) that is comprised of six A > P (red asterisks), five P > A, and six immature HCs (yellow and white arrows, (C)). (G–J) Selected optical sections of (D–F). (K–R) An afferent neuron (M) contacts six Emx2-negative P > A HCs ((K), asterisks, (N-R), circles) and one immature HC ((N), white arrow; (R), arrowhead) in a *myo6b:actb1-GFP* neuromast that is comprised of six A > P, six P < A (red asterisks) and two pairs of immature HCs (yellow and white arrows, (K)). Yellow and white arrowheads in (I) and (R) repectively, point to nerve fibers associating with immature HCs. Total sample number for (C–J) and (K–R) = 5. Scale bar = 10 μm.

DOI: https://doi.org/10.7554/eLife.35796.002

HGn39D transgenic background, in which all afferent neurons innervating each neuromast are GFP-positive (*Nagayoshi et al., 2008*; *Faucherre et al., 2009*). We then injected these embryos with *neuroD:tdTomato* to label individual afferent neurons. This enabled us to examine the innervation pattern of single neurons in the context of the entire afferent processes beneath each neuromast. In wildtype neuromasts, fibers of a single-labeled *neuroD:tdTomato*-positive neuron only overlapped with a subset of the HGn39D-GFP-positive afferents and ramified approximately one half of the neuromast (*Figure 2A–D*). By contrast, *emx2* ko neuromasts, in which all HCs are oriented towards the anterior direction (*Figure 2E,I*, P > A), showed an altered neuronal branching pattern (*Figure 2F–H, J–L*). The majority of the single neurons in *emx2* ko neuromasts showed a broader distribution (*Figure 2G*), when compared to controls (*Figure 2B–D*). Despite these broader distributions, there were always some GFP-positive nerve endings that did not overlap with tdTomato signals (*Figure 2F–H*, circles). Additionally, a smaller proportion of single fibers overlapped with less than half of the GFP-positive nerve endings in *emx2* ko neuromasts (*Figure 2J–L*), when compared to controls (*Figure 2B–D*). We quantified the average percentage of overlap between tdTomato- and GFP-positive areas and found that in wildtype fish, the overlap by a single neuron is consistently around 50% (*Figure 2U*, *Figure 2—source data 1*). In contrast, single neurons in *emx2* ko clearly segregated into two populations and showed either dramatically more (70%) or less (20%) overlap than the wildtype neurons (*Figure 2U*, p=0.0016 for *emx2* ko major, p=0.0012 for *emx2* ko minor). Based on these measurements, we named the two populations major and minor axons, respectively (*Figure 2U*). Similar major (*Figure 2N–P,U*, p=0.0006) and minor (*Figure 2R–T,U*, p=0.0005) branching patterns were observed for *emx2* gof neuromasts, in which all HCs are oriented in the A > P direction (*Figure 2M,Q*). The altered branching pattern of a single afferent neuron in both *emx2* ko and gof mutants suggests that in addition to establishing hair bundle orientation, Emx2 also regulates the pattern of neuronal innervation.

## Both major and minor afferent neurons contact HCs in *emx2* mutants

To investigate whether the changes in the branching pattern of individual neurons in *emx2* mutants represent a change in the number of HCs innervated, we conducted similar single-neuron labeling experiments using the *myo6b:actb1-GFP* transgenic line to more definitively correlate the relationship between nerve endings and HC bodies. We defined nerve endings that juxtaposed the basal region of HCs as contacts between the HC and the afferent. *Figure 3A–D3* illustrates a wildtype neuromast, in which a single afferent contacts all the HCs that are in a P > A orientation (*Figure 3A–C*, 4 out of 9 total HCs, *Figure 3D-D3*, arrowheads, *Table 1*, 52.9%, *Table 1—source data 1*). Similar contacts between nerve endings and HCs were found in *emx2* mutants (*Figure 3E-T2*). In *emx2* ko, major afferent neurons contacted more than half (*Figure 3E–H3*, 8 out of 10 total HCs, *Table 1*, 74.8%, p=5.19E-20, *Table 1—source data 1*) while minor afferent neurons contacted less than half of the total number of HCs within a neuromast (*Figure 3I–L3*, 4 out of 13 HCs, *Table 1*, 25.4%, p=1.41E-21, *Table 1—source data 1*). Similar to *emx2* ko afferents, major and minor neurons contacted a significantly more and fewer HCs respectively in *emx2* gof neuromasts compared to wildtype afferents (Example major, *Figure 3M-P3*, 10 out of 12 HCs, arrowheads, *Table 1*, 75.2%, p=1.82E-25; example minor, *Figure 3Q-T2*, 3 out of 11 HCs, arrowheads, *Table 1*, 25.8%, p=1.08E-18, *Table 1—source data 1*). However, in *emx2* mutants, the nerve endings of both major and minor neurons were juxtaposed to HC bodies (*Figure 3H1–H3, L1–L3, P1–P3, T1–T2*, arrowheads) similar to controls (*Figure 3D1–D3*), suggesting that they were successfully making HC contacts. We also examined the frequency of major and minor single afferent neurons obtained using single fiber

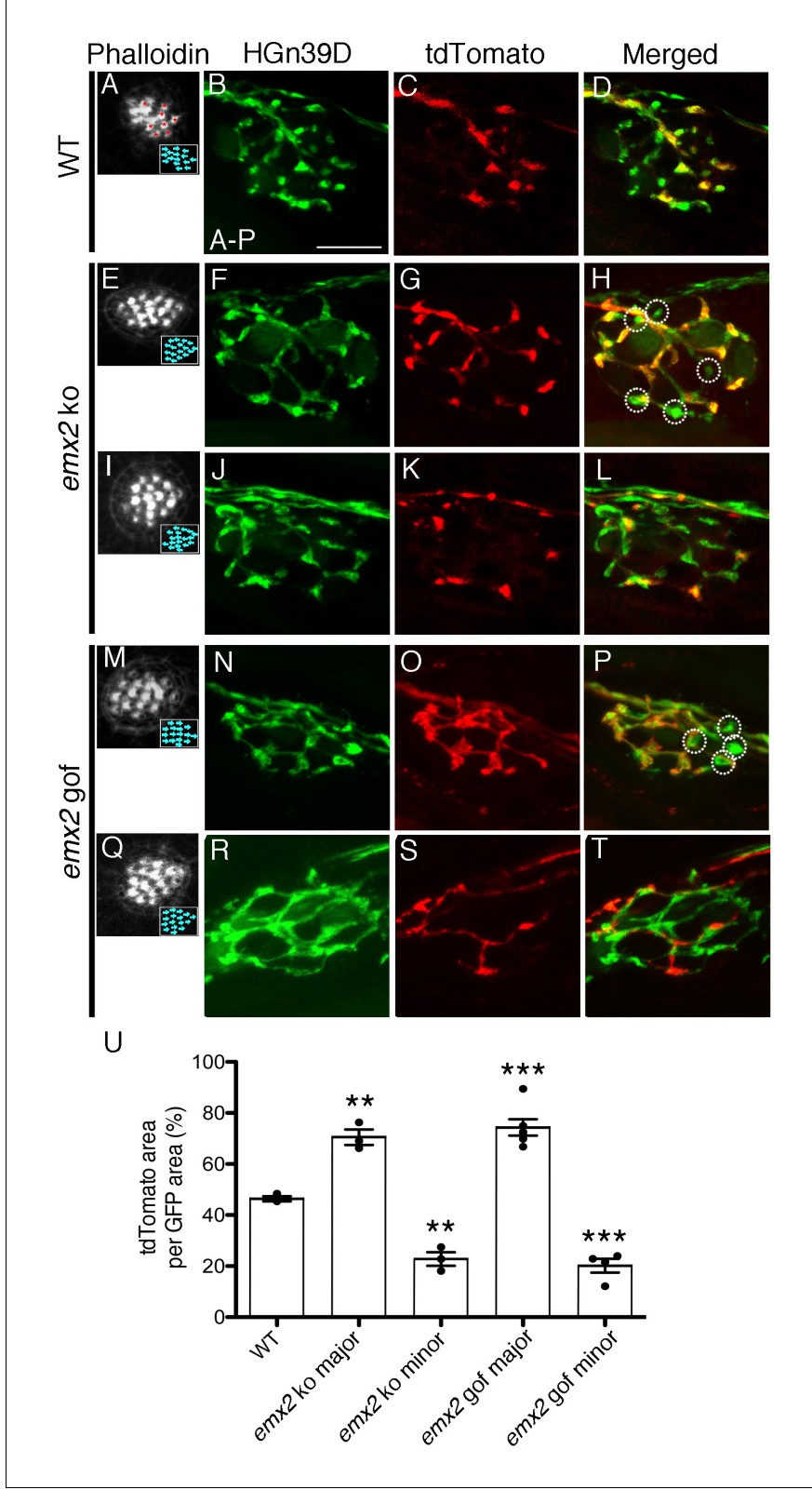

**Figure 2.** The branching pattern of a single afferent neuron in *emx2* mutant neuromasts is altered. (**A–D**) A wildtype neuromast showing the seven A > P and eight P > A (red asterisks) HCs. All neurons are labeled green (HGn39D), while a single neuron is labeled red (*neuroD:tdTomato*). The fibers of the single *tdTomato*-positive neuron (**C**) overlap with a subset of the GFP-positive neurons (**B**) in the posterior part of neuromast (**D**). (**E–L**) Two

*Figure 2 continued on next page*

*Figure 2 continued*

*emx2* ko neuromasts that contain only P > A HCs are shown (**E,I**). (**E–H**) The overlap between the single tdTomato-positive (**G**) and the GFP-positive neurons (**F**) in the neuromast is much broader than in wildtype (**D**). By contrast, the representation of tdTomato signals within the GFP-positive fibers of another *emx2* ko neuromast (**I–L**) is less than the one in wildtype (**D**). (**M–T**) Both *emx2* gof neuromasts (**M, Q**) show only A > P HCs, but one afferent neuron shows a broader distribution (**M–P**), whereas the other shows more restricted distribution of tdTomato in the neuromast (**Q–T**), when compared to the wildtype (**D**). Circles in (**H,P**) show fibers that are only GFP-positive. (**U**) Percentages of tdTomato-positive (single neuron) area per total GFP-positive (all neurons) area. The number of neuromasts: WT n = 3; *emx2* ko major n = 3, minor n = 3; *emx2* gof major n = 6, minor n = 4, the one-way ANOVA was used for the comparisons to WT. **p<0.01, ***p<0.001. Scale bar = 10 μm.
DOI: https://doi.org/10.7554/eLife.35796.003
The following source data is available for figure 2:

**Source data 1.** Percentages of tdTomato-positive area per total GFP-positive area.
DOI: https://doi.org/10.7554/eLife.35796.004

labeling and found they were in an approximate respective ratio of 3:2, for both *emx2* ko and gof neuromasts (*Table 1*). Overall these results indicate that in *emx2* mutants both major and minor neuronal fibers make contacts with HCs that are similar to those found in wildtype. Based on these similarities, it is likely that these contacts in *emx2* mutants may represent synapses.

## Both major and minor afferent nerve endings are associated with presynaptic specializations in *emx2* mutants

To determine whether all afferent nerve endings form synapses with HCs in *emx2* mutants, we analyzed *emx2* mutants using an antibody to a pre-synaptic protein, RibeyeB. In wildtype larvae, afferent fibers are associated with presynaptic RibeyeB staining (*Figure 4B–D3*). A single afferent neuron appears to innervate all the A > P or P > A HCs in the neuromast, and afferents are associated with RibeyeB-labeled presynapses (Example A > P, *Figure 4A,C–D*). In both *emx2* ko and gof neuromasts, we observed that major as well as minor afferent neurons associated with RibeyeB-labeled presynapses (*Figure 4E–T3*, white arrows). However, we also observed both major and minor afferent nerve endings that were adjacent to presynapses but failed to directly associate (*Figure 4E–T3*, white arrowheads). When we quantified the number of RibeyeB puncta that were colocalized with tdTomato-positive nerve endings per HC, we found that this number was not changed for the major fibers, but significantly decreased for the minor fibers for both loss- and gain-of-emx2 function neuromasts (*Figure 4—source data 1*, p=0.069 for *emx2* ko major, p=0.007 for *emx2* ko minor, p=0.383 for *emx2* gof major, p=0.002 for *emx2* gof minor). We also quantified the degree of colocalization or the proportion of presynaptic RibeyeB label that overlapped with the afferent-fiber label using Mander's colocalization analysis. Consistent with fewer RibeyeB puncta associating with afferent fibers, the Mander's colocalization coefficients were also decreased in minor fibers of *emx2* mutants compared to wildtype (*Figure 4—source data 2*, p=0.159 for *emx2* ko major, p=0.048 for *emx2* ko minor, p=0.011 for *emx2* gof major, p=0.037 for *emx2* gof minor). Together, these results suggest that a majority of the afferent nerve endings in *emx2* mutants and wildtype are associated with presynapses. However, the presence of nerve endings juxtaposed to HC bodies that do not associate with presynapses (incomplete synapses) are more prevalent in the *emx2* mutants than in wildtype (*Sheets et al., 2017*), primarily in the minor fibers.

## Incomplete synapses are present in *emx2* mutant neuromasts

To further assess synapse formation, we examined RibeyeB immunoreactivities in conjunction with the postsynaptic protein, Maguk (membrane associated guanylate kinases), in wildtype and *emx2* mutants. Previous work has demonstrated that this co-label can reliably be used to quantify complete and incomplete synapses in neuromasts (*Sheets et al., 2011*; *Sheets et al., 2017*). In wildtype neuromasts (*Figure 5A–D2*), the majority of RibeyeB and Maguk puncta were juxtaposed with each other, and based on this co-label, we observed on average three complete synapses per HC (*Figure 5M*; *Sheets et al., 2017*). Similar numbers of complete synapses per HC in both *emx2* ko and gof mutants were observed (*Figure 5E–H2, I–L2, M*, p=0.103 for *emx2* ko, p=0.878 for *emx2* gof, *Figure 5—source data 1*). We also used Mander's colocalization analysis to quantify the

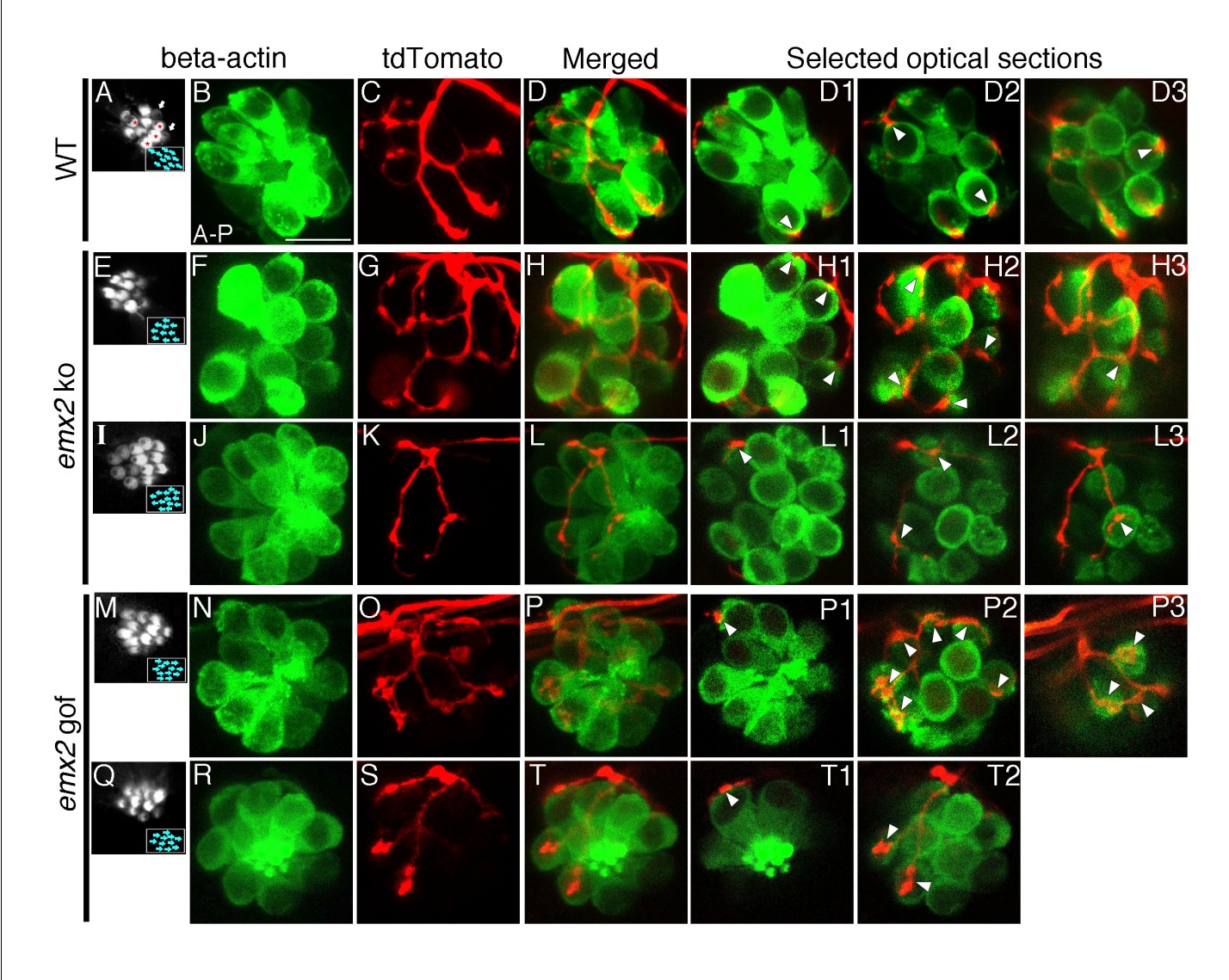

**Figure 3.** The number of HCs innervated by a single neuron is altered in *emx2* mutants. (**A–D3**) A wildtype neuromast (**A**) showing five A > P, four P > A (red asterisks), and two immature HCs (white arrows). A single *neuroD:tdTomato*-labeled afferent neuron (**C**) contacts four P > A *myo6b:actb1-GFP* HCs (red asterisks). (**D1–D3**) Selected optical sections showing nerve fibers (red) contacting cell bodies (green) of HCs (white arrowheads, 4/9 HCs). (**E–H3**) A single afferent neuron (**G**) contacting eight out of ten total P > A HCs (**H–H3**) in an *emx2* ko neuromast (arrowheads). (**H1–H3**) Selected optical sections of (**H**). (**I–L3**) A single labeled afferent neuron (**K**) contacts four out of thirteen total HCs in an *emx2* ko neuromast (arrowheads). (**L1–L3**) Selected optical sections of (**L**). (**M–T**) A single afferent neuron (**O**) contacts ten out of twelve A > P Gfp-positive HCs (**M,N**) in an *emx2* gof neuromast (arrowheads). (**P1–P3**) Selected optical sections of (**P**). (**Q–T2**) A single afferent neuron (**S**) contacts three out of eleven total A > P HCs (**Q, R,T**) in an *emx2* gof neuromast (arrowheads). (**T1–T2**) Selected optical sections of (**T**). The number of neuromasts: WT, n = 11; *emx2* ko major, n = 8, minor, n = 4; *emx2* gof major n = 12, minor n = 6. Scale bar = 10 µm.

DOI: https://doi.org/10.7554/eLife.35796.007

proportion of RibeyeB label that overlapped with Maguk label. Using this analysis, we found no difference in *emx2* mutants, compared to wildtype (***Figure 5—source data 2***).

Although there is no difference in complete synapse number and overall pre-and post-synaptic colocalization, *emx2* mutant neuromasts showed an increase number of both total RibeyeB and Maguk per HC (p=6.09E-05 for RibeyeB of *emx2* ko, p=2.09E-04 for RibeyeB of *emx2* gof, p=3.35E-04 for Maguk of *emx2* ko, p=0.567 for Maguk of *emx2* gof, ***Figure 5—source data 3***). This was due to an increase in individual RibeyeB and Maguk puncta that were not juxtaposed with each other (***Figure 5H1–H2, L1–L2***, white and yellow arrows). We quantified the percentage of solitary RibeyeB

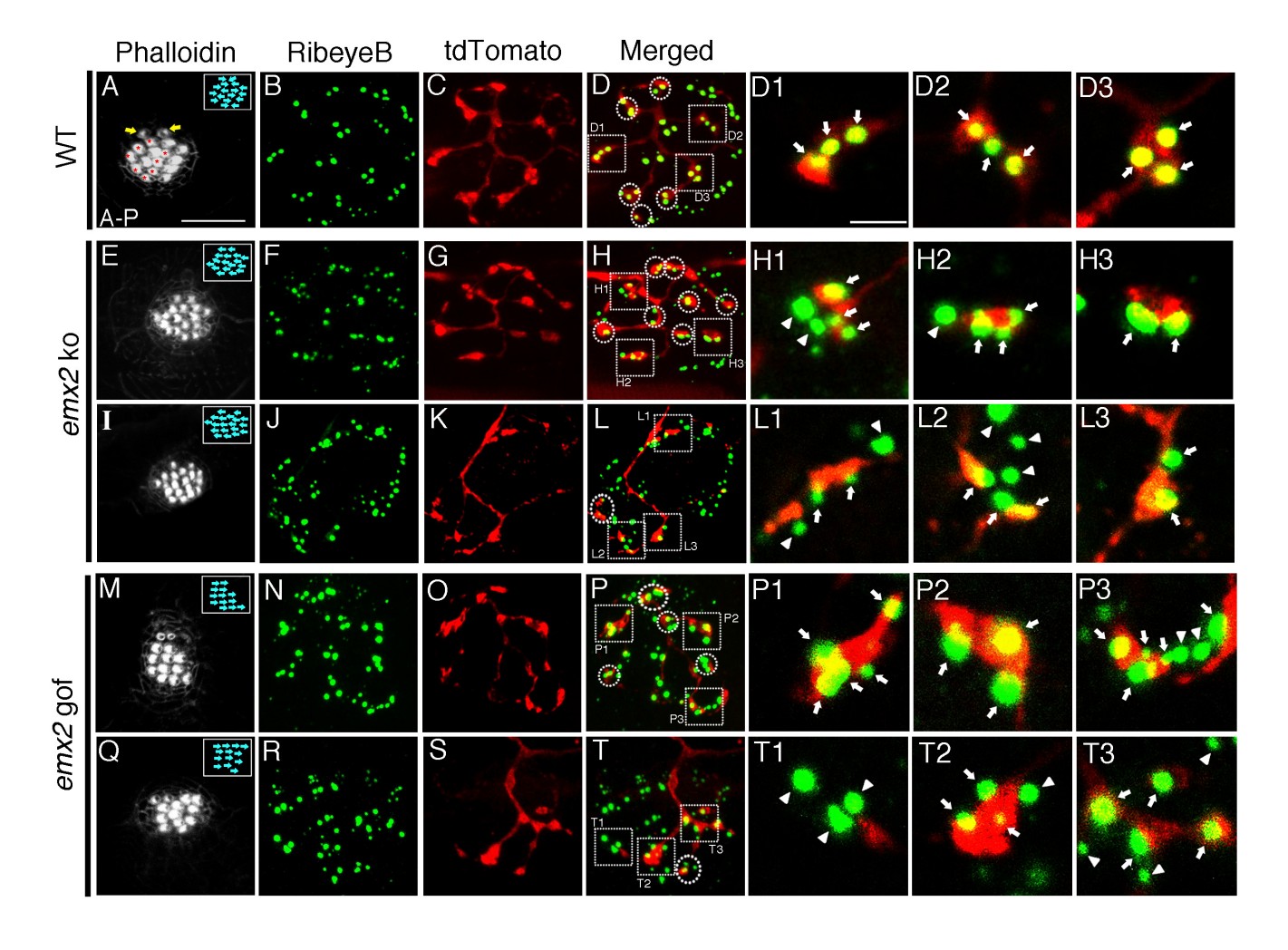

**Figure 4.** Association of single-labeled neurons with RibeyeB in *emx2* mutants. (**A–D**) A wildtype neuromast with eight A > P (red asterisks), seven P > A, and two immature HCs (yellow arrows). (**A**) This neuromast shows a single *neuroD:tdTomato* labeled afferent neuron (**C**) that contacts only A > P HCs (red asterisks, **A**) that is stained with anti-RibeyeB antibody (**B**). (**D**) Merged labels of nerve endings and RibeyeB. (**D1–D3**) Expansion of square insets in (**D**) from selected 1 μm optical sections. (**E–H3**) A major *neuroD:tdTomato* afferent neuron (**G**) in an *emx2* ko neuromast consisting of only P > A HCs (**E**) that is stained with RibeyeB antibody (**F**). (**H**) Merged labels of nerve endings and Ribeye. (**H1–H3**) Expansion of square insets of (**H**) from selected 1 μm optical sections. (**I–L3**) An *emx2* ko neuromast (**I**) with a single minor *neuroD:tdTomato*-labeled afferent neuron (**K**) stained with RibeyeB antibody (**J**). (**L**) Merged labels of nerve endings and RibeyeB. (**L1–L3**) Expansion of square insets in (**L**) from selected 1 μm optical sections. (**M–P3**) A major afferent neuron (**O**) in an *emx2* gof neuromast contacts only A > P HCs (**M**) stained wiht RibeyeB antibody (**N**). (**P**) Merged labels of nerve endings and RibeyeB. (**P1–P3**) Expansion of square insets in (**P**) from selected 1 μm optical sections. (**Q–T3**) A minor afferent neuron (**S**) in an *emx2* gof neuromast that is stained with RibeyeB antibody (**R**). (**T**) Merged labels of nerve endings and RibeyeB. (**T1–T3**) Expansion of square insets in (**T**) from selected 1 μm optical sections. Squares and circles in (**D,H,L,P,T**) indicated all the HCs with RibeyeB label that are contacted by each fiber. White arrows and arrowheads show Ribeye puncta that are juxtaposed to labeled nerve endings or not, respectively. The number of neuromasts: WT, n = 8; *emx2* ko major, n = 6, minor, n = 4; *emx2* gof major n = 8, minor n = 7. Scale bar in (**A**) equals 10 μm and applies to all lower magnification images, and scale bar in (**D1**) equal 2.5 μm and applies to all higher magnification images.

DOI: https://doi.org/10.7554/eLife.35796.008

The following source data is available for figure 4:

**Source data 1.** Quantification of colocalization between RibeyeB and tdTomato-positive nerve endings per HC.
DOI: https://doi.org/10.7554/eLife.35796.009
**Source data 2.** Mander's coefficient of RibeyeB and tdTomato colocalization.
DOI: https://doi.org/10.7554/eLife.35796.010

**Table 1.** The percentages of innervated HCs by major and minor afferent neurons in *emx2* mutants

| | WT | *emx2* ko major | *emx2* ko minor | *emx2* gof major | *emx2* gof minor |
|---|---|---|---|---|---|
| Innervated HCs/total HCs (%) | 52.9 ±2.3 | 74.8* ±5.1 | 25.4* ±4.4 | 75.2* ±4.5 | 25.8* ±7.3 |
| Sample number (N) | 22 | 17 | 11 | 26 | 17 |

*p<0.001 compared to WT by one-way ANOVA
DOI: https://doi.org/10.7554/eLife.35796.006
The following source data available for Table 1:
Source data 1. Percentages of innervated HCs by single labeled afferent neuron.
DOI: https://doi.org/10.7554/eLife.35796.005

or Maguk puncta per neuromast and found that there were significantly more solitary puncta in both *emx2* ko and gof mutants compared to controls (*Figure 5H–H2,L–L2, N*, p=5.05E-07 for RibeyeB of *emx2* ko, p=1.35E-08 for RibeyeB of *emx2* gof, p=9.98E-09 for Maguk of *emx2* ko, p=0.033 for Maguk of *emx2* gof, *Figure 5—source data 4*). The uncoupling of pre- and post-synaptic components support that idea that in addition to forming authentic synapses, *emx2* mutants also have failures in synapse formation. Therefore, we asked whether afferent neurons in *emx2* mutants are functionally active, by measuring neuronal responses during hair bundle displacements in *emx2* mutants.

## Afferent neurons in *emx2* mutants show predicted directional selectivity

To determine whether *emx2* mutant afferent neurons are functional, we used in vivo calcium imaging and electrophysiology to measure afferent neuron activity. We used a waterjet to directionally stimulate HCs within a neuromast in either an anterior or posterior direction while simultaneously imaging postsynaptic calcium signals or recording spikes.

We used GCaMP6s to image calcium signals in the afferent process directly beneath the neuromast HCs during a 500 ms waterjet stimulation (*Figure 6*). In wildtype, we observed robust calcium signals in the innervating afferent process during both A > P and P > A directed waterjet stimulation (*Figure 6A,A'*). In wildtype afferents, calcium responses to each directional stimulation (A > P or P > A) occurred in distinct domains of the afferent process (*Figure 6A,A'*). In contrast, calcium signals were only observed in response to P > A stimulation in *emx2* ko afferents (*Figure 6B'*). Inversely, calcium signals were only detected in response to A > P stimulations in *emx2* gof afferents (*Figure 6C*). No afferent calcium signals were observed in *emx2* ko mutants during A > P stimuli (*Figure 6B*) or in *emx2* gof mutants during P > A stimuli (*Figure 6C'*). These results indicate that HCs and afferent neurons in *emx2* mutants are functional and directionally selective. Our results are also consistent with the single-directional sensitivity of the *emx2* gof (A > P) hair bundles compared to wildtype hair bundles in both A > P and P > A directions (*Jiang et al., 2017*). We quantified the magnitude of the calcium signal in the entire afferent process for each stimulus. We found that for the A > P stimulus, the *emx2* gof calcium signals were 41% higher compared to wildtype (*Figure 6D,D'*, p=0.027, *Figure 6—source data 1*). Similarly, in *emx2* ko fish, the P > A calcium signals were 58% higher compared to wildtype (*Figure 6E,E'*, p=0.0003, *Figure 6—source data 1*). These results suggest that compared to wildtype, in *emx2* mutants, afferents are able to recruit additional HCs to respond to their respective direction of stimulus.

To investigate whether there are differences in the synaptic transfer between individual HCs of wildtype and mutants, we quantified the magnitude of the afferent calcium signals per HC. To identify HCs within the neuromast, we used FM 4–64 to label the HCs after afferent calcium imaging (*Figure 6A'',B'',C''*). Then we measured the magnitude of calcium signals contained within 3 μm regions of interest (ROIs) on the afferent process beneath each functional HC (*Figure 6A'',B'',C''*). Using this approach, we found that per HC, the magnitude of the afferent calcium signal was the same compared to wildtype (*Figure 6F,G*, A > P direction: GCaMP6 ΔF/F, wildtype 70.5%, *emx2* ko 71.6%, p=0.99; P > A direction: GCaMP6 ΔF/F, wildtype 51.6%, *emx2* gof 73.4%, p=0.48, *Figure 6—*

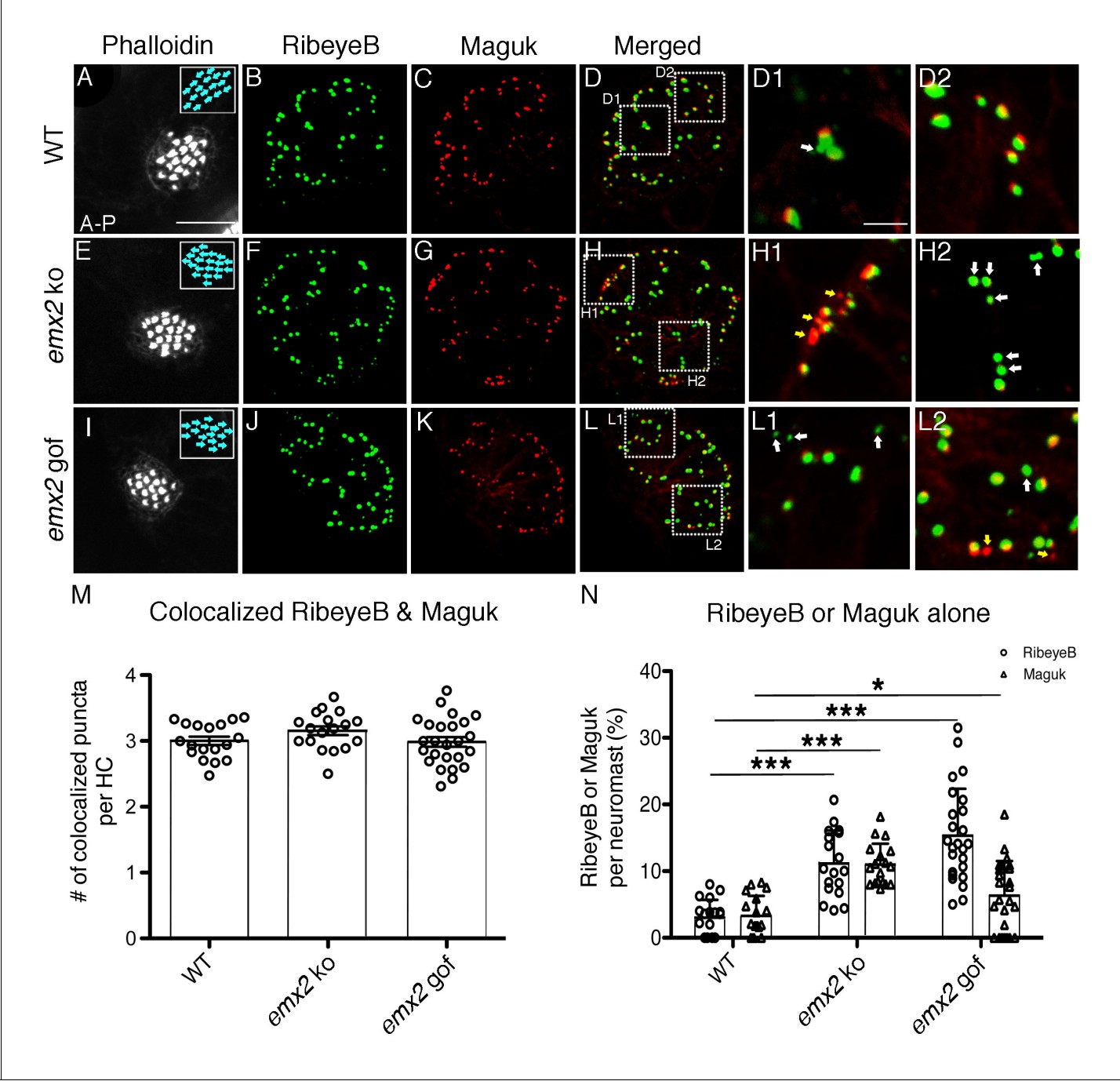

**Figure 5.** The number of RibeyeB- and Maguk-positive puncta is altered in *emx2* mutant neuromasts. (A–D) A wildtype neuromast consisting of nine A > P and eight P > A HCs is stained with phalloidin (A), anti-RibeyeB (B) and anti-Maguk (C) antibodies. (D) A merged image of (B) and (C) showing co-localization of RibeyeB and Maguk. (D) A merged image of (B) and (C) showing co-localization of RibeyeB and Maguk. (D1,D2) Insets in (D) from selected optical sections showing the co-localized RibeyeB and Maguk but one has RibeyeB staining only (white arrow). (E–H2) An *emx2* ko neuromast with only P > A HCs (E) is stained with anti-RibeyeB (F) and anti-Maguk (G) antibodies. (H) A merged image of (F) and (G). (H1,H2) Insets in (H) from selected optical sections showing RibeyeB (white arrows) or Maguk (yellow arrows) staining alone. (I–L2) A phalloidin-labeled *emx2* gof neuromast with only A > P HCs (I) is stained with anti-RibeyeB (J) and anti-Maguk (K) antibodies. (L) A merged image of RibeyeB and Maguk. (L1,L2) Insets in (L) from selected optical sections showing staining of RibeyeB (white arrows) or Maguk (yellow arrows) alone. (M) The number of colocalized RibeyeB and Maguk puncta per HC is similar between wildtype and *emx2* mutant neuromasts. (N) The percentages of single RibeyeB and Maguk puncta per neuromast are higher in *emx2* mutant than wildtype neuromasts. The number of neuromasts: WT n = 18, *emx2* ko n = 18, *emx2* gof n = 25, obtained

*Figure 5 continued on next page*

*Figure 5 continued*

from three independent experiments. The one-way ANOVA was used for the comparisons shown in (M) and (N). *p<0.05, ***p<0.001. Scale bar in (A) equals10 μm and applies to all lower magnification images, and scale bar in D1 equal 2.5 μm and applies to all higher magnification images.

DOI: https://doi.org/10.7554/eLife.35796.011

The following source data is available for figure 5:

**Source data 1.** Quantification of colocalized RibeyeB and Maguk puncta per HC.

DOI: https://doi.org/10.7554/eLife.35796.012

**Source data 2.** Mander's coefficient of RibeyeB and Maguk colocalization.

DOI: https://doi.org/10.7554/eLife.35796.013

**Source data 3.** Quantification of total number of RibeyeB and Maguk puncta per HC.

DOI: https://doi.org/10.7554/eLife.35796.014

**Source data 4.** Percentages of solitary RibeyeB and Maguk puncta per neuromast.

DOI: https://doi.org/10.7554/eLife.35796.015

---

*source data 2*). This suggests that the overall properties of each HC-afferent contact are not dramatically altered in *emx2* mutants.

Although calcium imaging provides an excellent overview of the response properties within the entire afferent process beneath an individual neuromast, our imaging approach does not have the sensitivity to detect and resolve the timing of single action potentials. Therefore, it is possible that there were evoked responses in the off -stimulus direction (ie: A > P direction in *emx2* ko) that we could not detect using calcium imaging. To address this possibility, we performed in vivo electrophysiological recordings from the afferent cell bodies in the posterior lateral line ganglion, while stimulating neuromast HCs with a waterjet.

In wildtype larvae, within each neuromast, each afferent neuron contacts multiple HCs with either an A > P or P > A directional sensitivity (*Figure 1B*; *Nagiel et al., 2008*; *Faucherre et al., 2009*). Consistent with this sensitivity pattern, during a 500 ms waterjet stimulus, we were able to record trains of action potentials (spikes) either during A > P or P > A stimuli in wildtype afferents (*Figure 7A,B*, wildtype, A > P, 5/15, 33%, P > A, 10/15, 66%, n = 15 afferent neurons, *Figure 7—source data 1*). When these same wildtype afferents were stimulated in the opposite directions (off-stimulus direction), the average spike number was dramatically reduced (*Figure 7B'*). When we recorded spikes in *emx2* mutant afferent neurons, we only identified P > A afferents in *emx2* ko mutants (n = 8/8 neurons; *Figure 7B*) and A > P afferents in *emx2* gof mutants (n = 7/7 neurons; *Figure 7B*). The number of spikes in the on-stimulus direction, in *emx2* mutants was not greater than wildtype (wildtype versus *emx2* ko, p=0.65; wildtype versus *emx2* gof, p=0.74). When *emx2* mutant afferents were stimulated in the off-stimulus direction, similar to wildtype afferents, the spike numbers were dramatically reduced (*Figure 7B'*). Together, our calcium imaging and electrophysiological results both indicate that the afferent neurons in *emx2* mutants are functional and respond in a predicted manner according to their hair bundle orientation.

## Selective neuronal innervation is independent of hair bundle orientation

Thus far, our results support the idea that Emx2 expressed in HCs is required for establishing directional selectivity of afferent neurons, in addition to determining hair bundle orientation (*Jiang et al., 2017*). Next, we asked whether it was hair bundle orientation per se or the identity of the HC conferred by Emx2 that is important to establish afferent selectivity. To address this question, we investigated a core planar cell polarity mutant, *vangl2^{m209}*, in which a recessive frameshift mutation in *vangl2* results in a functional null protein (*Solnica-Krezel et al., 1996*; *Jessen et al., 2002*). Hair bundles in neuromasts of *vangl2^{m209}* are misoriented due to misalignment of the basal body and in turn the associated kinocilium, which directs bundle orientation during development (*Figure 8E,M,P*; *López-Schier and Hudspeth, 2006*). Although *emx2* mutants have misoriented hair bundles in both the inner ear and neuromasts, the orientations are not random as in the *Vangl* mutants. Importantly in both mutants, the misoriented hair bundles are attributed to mis-localization of the kinocilium (*Jiang et al., 2017*). Based on the implication of the mis-located basal body/kinocilium in both of these mutants, we asked whether the neuronal innervation is affected in *vangl2^{m209}*. First, we investigated whether Emx2 is expressed in *vangl2^{m209}*. Our results showed that although hair bundle

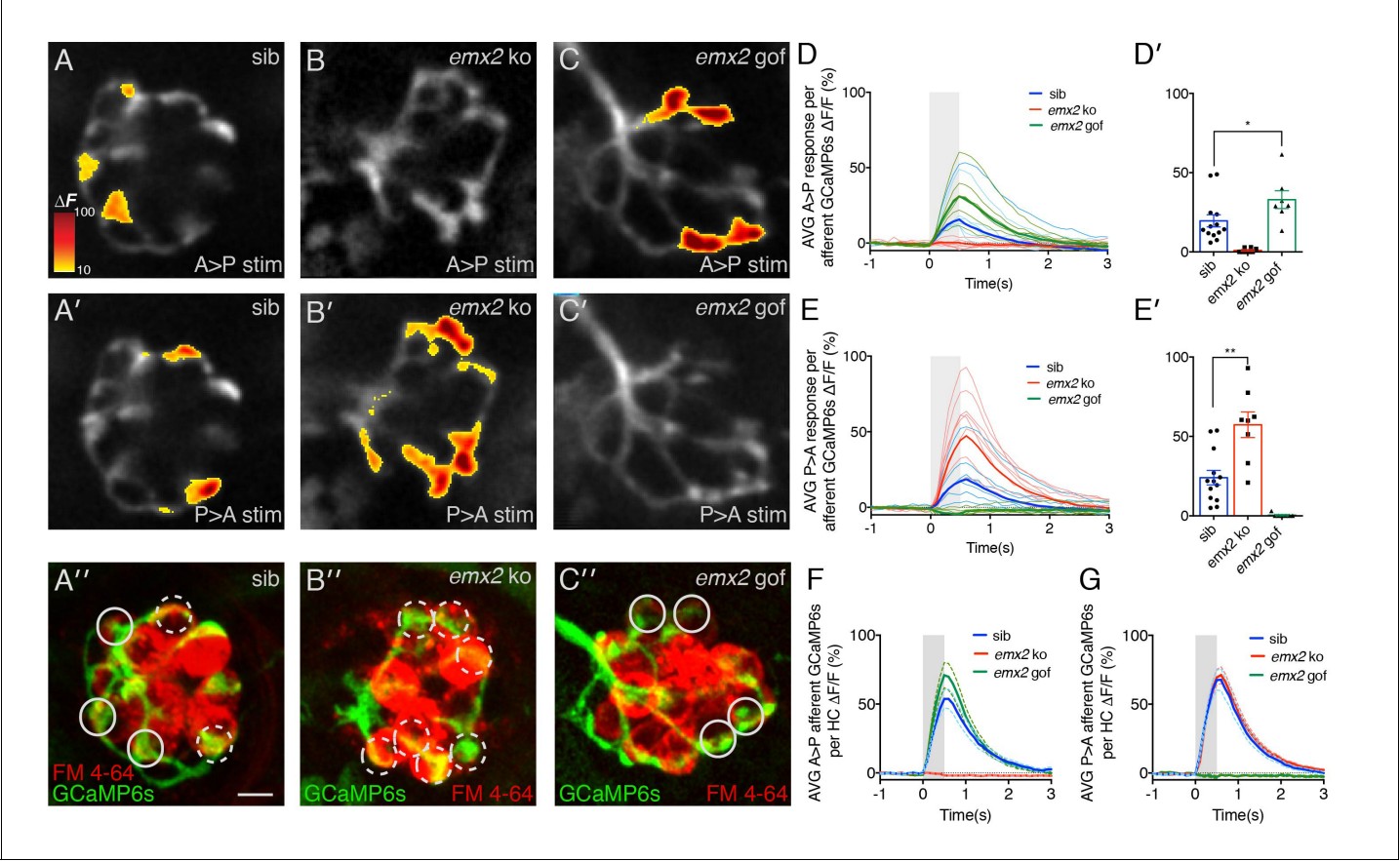

**Figure 6.** Calcium imaging in afferent process of wildtype and *emx2* mutant neuromasts. (A, A′) Representative calcium signals during a A > P (A) and P > A (A′) stimulation in an individual, wildtype sibling afferent process. (B,B′) Afferent calcium signals are detected in *emx2* ko mutants in response to P > A (B′), but not in response to A > P (B) stimulation. (C,C′) Afferent calcium signals are only detected in *emx2* gof mutants in response to A > P (C), but not in response to P > A (C′) stimulations. (A″,B″,C″) The same afferent processes (green) as the panels above, except the HCs are labeled with FM 4–64 (red). (D–E′) Quantification of the mean response per afferent process in wildtype and *emx2* mutants. Compared to wildtype, afferent processes in *emx2* gof mutants have more signal during A > P stimulation (D, D′), and afferent processes in *emx2* ko mutants during P > A stimuli (E, E′), n = a minimum of 7 neuromasts. (F,G) Quantification of the afferent calcium signals per HC. Circular ROIs in A″, B″ and C′ show representative ROIs used to quantify calcium signals beneath HCs. Solid and dashed circles represent afferent foci responsive in either the A > P or P > A direction, respectively. For A > P stimuli the average afferent calcium response per HC is not different between wildtype and *emx2* gof (F). Similarly, for P > A stimuli, the average afferent calcium response per HC is not different between wildtype and *emx2* ko (G). Calcium signals are colorized according to the ΔF heat map and superimposed onto a baseline GCaMP6s image. The number of neuromasts: WT, n = 13; *emx2* ko, n = 8; *emx2* gof n = 7, obtained from two independent experiments. A one-way ANOVA was used for the comparison in (D′) and (E′). A Kruskal-Wallis test was used for comparisons in (F) and (G), *p<0.05, **p<0.01. Scale bars = 5 μm.

DOI: https://doi.org/10.7554/eLife.35796.016
The following source data is available for figure 6:

**Source data 1.** Magnitude of afferent calcium signals per Neuromast.
DOI: https://doi.org/10.7554/eLife.35796.017
**Source data 2.** Magnitude of afferent calcium signals per HC.
DOI: https://doi.org/10.7554/eLife.35796.018

orientation is random in *vangl2*[m209] mutants, Emx2 is still expressed in half of the HCs within a neuromast (*Figure 8E–H2*, example 4/8 HCs, *Figure 8I*, p=0.67, *Figure 8—source data 1*), similar to controls (*Figure 8A–D2*, example 6/12 HCs, *Figure 8I*). The maintenance of 50% of Emx2-positive HCs in *vangl2*[m209] mutants suggests that hair bundle misorientation in this mutant is independent of *emx2* (*Figure 8I*). Furthermore, the hair bundle orientation of double mutants of *vangl2*[m209] with *emx2* ko or gof was also random (*Figure 8N,O*). The orientation defects in the double mutants resembled defects in *vangl2*[m209] (*Figure 8M*), rather than the unidirectional orientation phenotype

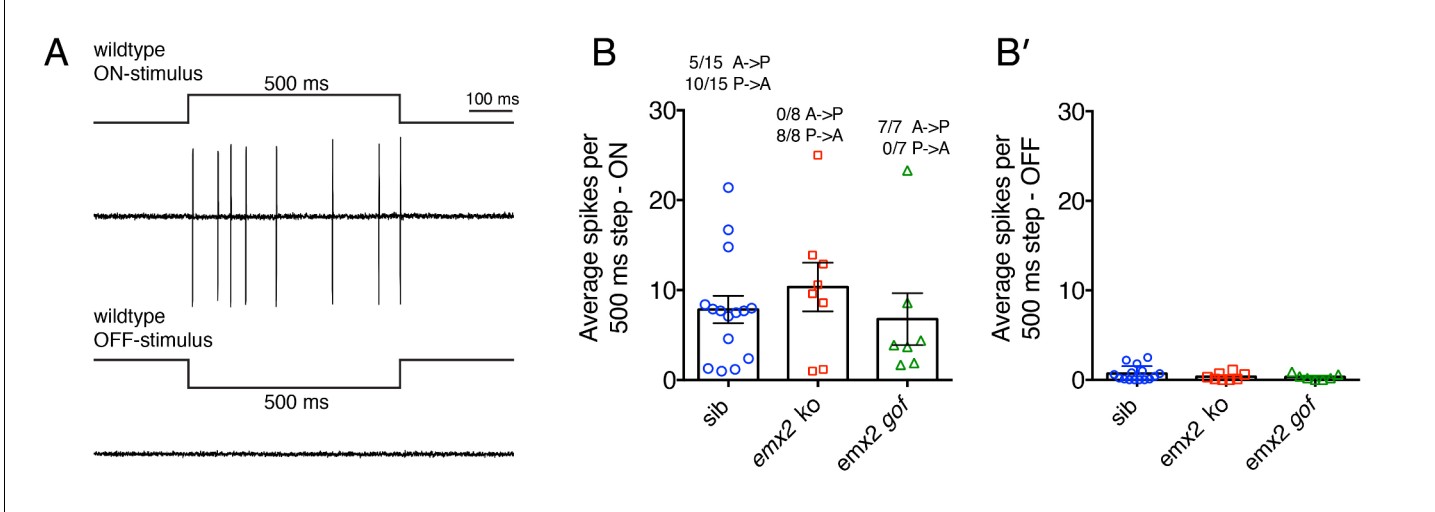

**Figure 7.** Electrophysiological recordings from wildtype and *emx2* mutant afferent neurons. (**A**) Example of wildtype spikes during a 500 ms ON and OFF waterjet stimulus. (**B**) Quantification of the average spikes per 10, 500 ms step stimuli in the direction of sensitivity (ON-stimulus). For wildtype, the A > P and P > A afferent neuron spike numbers were pooled. A similar number of spikes were detected in wildtype, *emx2* ko and *emx2* gof neurons. (**B'**) Quantification of the average spikes for the same neurons as in (**B**), except in the opposite direction (OFF-stimulus). The number of afferents recorded: WT, n = 15; *emx2* ko, n = 8; *emx2* gof n = 7, obtained from a minimum of three independent experiments. The one-way ANOVA was used for the comparison in (**B**) and (**B'**).

DOI: https://doi.org/10.7554/eLife.35796.019

The following source data is available for figure 7:

**Source data 1.** Average spikes per 10, 500 ms step stimuli.

DOI: https://doi.org/10.7554/eLife.35796.020

observed in *emx2* ko and gof single mutants (*Figure 8K,L*). Therefore, these genetic analyses indicate that *vangl2* is epistatic to *emx2*.

The above results demonstrate that the *vangl2* mutant is a good model to address the question whether hair bundle orientation per se is coupled to neuronal selectivity since hair bundle misorientation in this mutant is independent of *emx2*. When we examined the innervation patterns of single afferent neurons in *vangl2^{m209}* neuromasts, we observed that Emx2-mediated afferent selectivity was preserved. For example, a single afferent neuron can selectively innervate Emx2-positive HCs (Example, *Figure 8P–U*, 5 out of 12 total HCs, circles). In addition, the ratio of the number of HCs innervated by a single neuron within a neuromast is similar to wildtype (*Figure 8Y*, p=0.26, *Figure 8—source data 2*). Importantly, single neurons were able to select and contact only the Emx2-positive HCs (*Figure 8T–X*), despite the misoriented hair bundles. These results indicate that neuronal innervation in neuromasts occurs independent of hair bundle orientation and is directed by Emx2.

## Discussion

### Exclusive or inclusive role of Emx2 in neuronal selectivity

In a wildtype neuromast, a single afferent neuron innervates only HCs with bundles that are oriented in the same direction, which constitute about half of the total number of HCs. In *emx2* ko and gof neuromasts, in which all HCs have uni-directionally oriented bundles, we found that one population of afferent neurons can innervate more than half of the total number of HCs. These results suggest that these neurons have the capacity to innervate additional HCs of similar orientation based upon their availability. Nevertheless, our results also suggest that there is an upper limit to this capacity since a single neuron was unable to innervate all the HCs within a ko or gof *emx2* mutant neuromast. Coincidently, we also observed a smaller portion of single neurons innervating less than half of the total number of HCs in the *emx2* mutant neuromasts (*Table 1*).

What determines whether a neuron forms a major versus a minor innervation pattern in the mutant neuromasts? At least two models are consistent with our observations. We refer to these models as 'exclusive' and 'inclusive' based on the role of Emx2 in determining neuronal selectivity. In the exclusive model, the transcriptional activity downstream of Emx2 in HCs is sufficient to direct afferent innervation and determine the directional selectivity of each neuron (*Figure 9*). In this model, there are no other cues within neuromasts or among the lateral line ganglion neurons that influence this decision, at least not initially. However, interaxonal cross-inhibition could occur after the initial contact with HCs that refines or maintains the segregated pattern (*Changeux and Danchin, 1976*; *Deppmann et al., 2008*; *Mizumoto and Shen, 2013*). In this scenario, major and minor afferents in *emx2* mutants could arise due to timing; axons arriving at the neuromast first may innervate many cells and become major afferents. In contrast, axon that arrive slightly later may find fewer available HCs to innervate and thus become minor afferents.

In the inclusive model, Emx2, though important, is not the only factor regulating neuronal selectivity (*Figure 9*). For example, both pre- and post-synaptic signaling may be important to determine neuronal selectivity. In this scenario, major and minor innervation patterns in the *emx2* mutants may arise from the fact that afferent neurons that innervate A > P HCs (A > P neurons) are molecularly distinct from those that innervate P > A HCs (P > A neurons). In support of this notion, neurons that innervate Emx2-positive and -negative HCs in the mammalian maculae of the inner ear are most likely different from each other because their central axons project to different regions of the brain (*Maklad et al., 2010*). Therefore, neurons that innervate Emx2-positive and negative HCs in the lateral line system may also be molecularly different, similar to the neurons of the mouse maculae. Additionally, ample evidence from other systems indicates that specific synapse formation often requires partnering between pre- and post-synaptic signaling molecules such as ephrin-eph, semaphorin-plexin and other adhesion molecules (for review see, *Sanes and Yamagata, 2009*; *Margeta and Shen, 2010*; *Williams et al., 2010*; *Yogev and Shen, 2014*). The selective innervation pattern of the lateral line afferent neurons may be analogous to other pre- and postsynaptic neuronal pairing, requiring a match between the HCs (expressing downstream effectors of Emx2 or the lack thereof) and respective molecular determinants from the postsynaptic afferent neurons. Moreover, the described interactions of cellular processes between nascent HCs and afferent nerve terminals of a neuromast during synaptogenesis also provide some morphological evidence that there could be specific HC-neuron coupling (*Dow et al., 2015*).

Under the above inclusive scenario, the P > A neurons in the *emx2* ko neuromasts identify molecular cues from an expanded number of P > A HCs and become major afferents, whereas the A > P neurons are unable to identify their molecular cues, since A > P HCs are absent (*Figure 9*). Without a molecular target, these A > P neurons either form only a few synapses with P > A HCs or fail to form any synapses at all and become minor afferents. Likewise, in the *emx2* gof neuromasts, the A > P neurons expand their innervation pattern to accommodate the increase in A > P HCs and become major afferents, while the P > A neurons in the *emx2* mutant form minor innervation patterns. Previous axotomy studies showed that when all axons innervating a single neuromast were severed except one, the single remaining neuron was able to expand its innervation to HCs of the opposite orientation (*Pujol-Martí et al., 2014*). As a result of this expanded innervation, the remaining neuron responded to hair bundle deflections from two directions instead of one. If we assume that there are indeed two types of neurons innervating a given neuromast, then the axotomy data taken together with our data on *emx2* mutants suggest that other permissive factor(s) in addition to Emx2 instruct synapse formation and can expand innervations beyond their intended targets. Nevertheless, the inclusive model is not limited to molecular differences between neurons and could include other determinants from HCs and/or supporting cells. Furthermore, under either the inclusive or exclusive model, we postulate that the main source of Emx2 for neuronal selectivity resides in the HCs because we cannot detect measurable *emx2* mRNA or protein in the lateral line ganglion using in situ hybridization or immunostaining.

## Neurons of *emx2* mutant neuromasts showed predicted directional selectivity

Despite the altered innervation pattern observed in the *emx2* mutant neuromasts, our calcium imaging and extracellular neuronal recording results indicate that these neurons respond in the predicted manner according to the orientation of the HCs that they innervate; neurons of the *emx2* ko

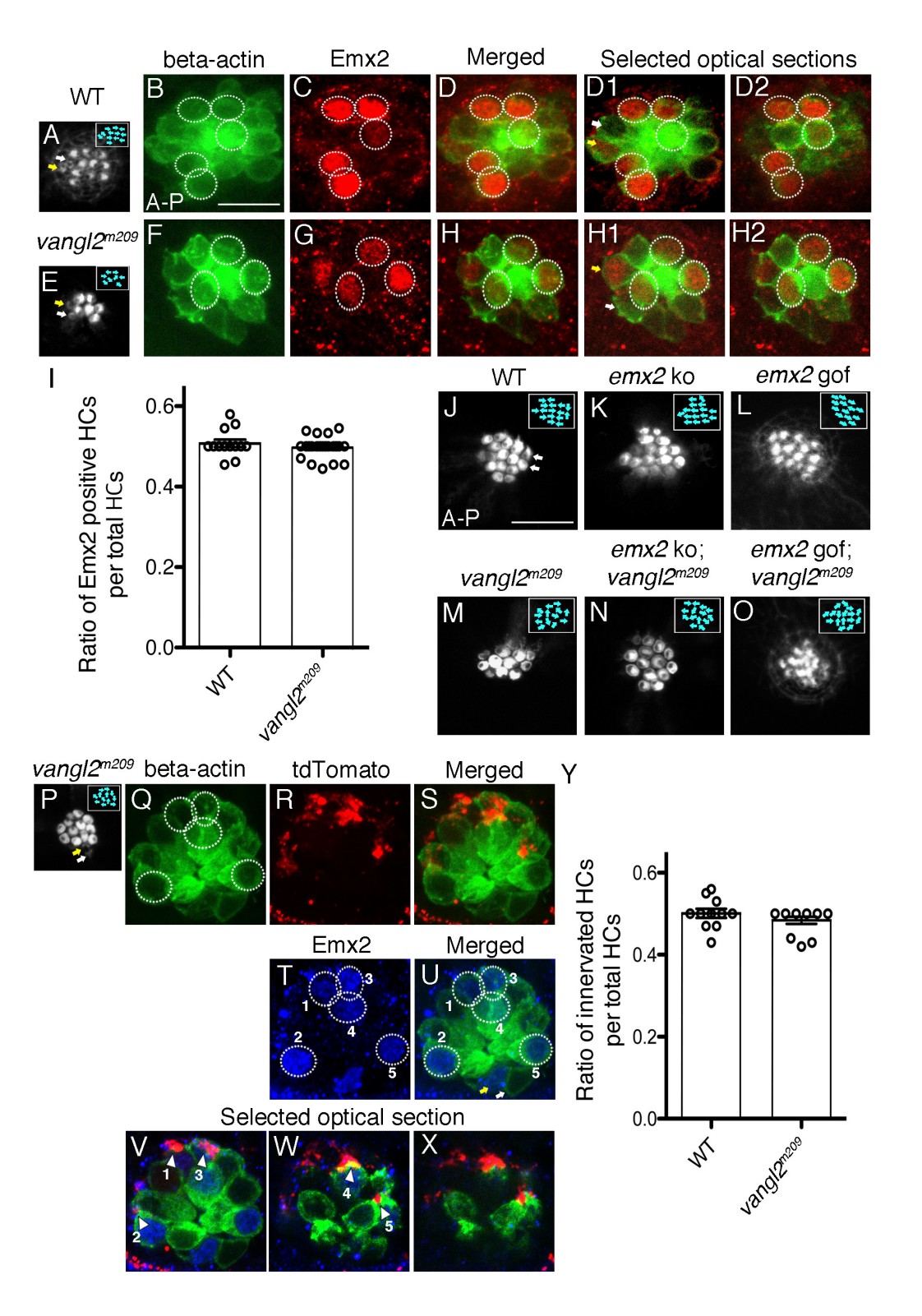

**Figure 8.** Selectivity of single-afferent neurons in *vangl2^m209* mutants correlates with Emx2 expression. (**A–D2**) A wildtype *myo6b:actb1-GFP* neuromast consists of five A > P, five P > A and two immature HCs (yellow and white arrows, (**A,D1**)), in which Emx2 expression is restricted to the five A > P (circles) and one of the immature HCs (yellow arrow, (**D1**)). (**D**) A merged image of (**B**) and (**C**), and (**D1,D2**) are selected optical sections of (**D**). (**E–H2**) A *vangl2^m209*; *myo6b:actb1-GFP* neuromast consists of six mature and two immature HCs (yellow and white arrows) that are randomly polarized (**E,F**), and

*Figure 8 continued on next page*

*Figure 8 continued*

Emx2 is expressed in three mature (circles) and one of the immature HCs (yellow arrow, (H1)). (H) A merged image of (E) and (F), and (H1,H2) are selected optical sections of (H). (I) The percentage of Emx2-positive HCs within a neuromast is similar between wildtype and *vangl2*^*m209*^ mutant neuromasts. N.S. Student's t-test. The number of neuromasts: WT, n = 10; *vangl2*^*m209*^, n = 22, combined results from three independent experiments. (J–O) Phalloidin-staining of neuromasts from various genotypes. (J) A wildtype neuromast containing seven A > P, seven P > A and two immature HCs (white arrows). Hair bundle orientation is random in *vangl2*^*m209*^ (M), *emx2* ko; *vangl2*^*m209*^ (N), and *emx2* gof; *vangl2*^*m209*^ (O) neuromasts. (P–X) Phalloidin (P) and GFP (Q) staining of a *vangl2*^*m209*^; *myo6b:actb1-GFP* neuromast showing random hair bundle orientation (n = 3). (R) Neuronal processes of a single *neuroD:tdTomato*-labeled afferent neuron contacts all five Emx2-positive HCs ((T), circles) in the neuromast (Q–X) and none of the Emx2-negative HCs. (V–X) Selected optical sections of (Q,R,T) showing the relationship between Emx2-positive HCs (blue nuclei in green HCs) and tdTomato-positive afferent processes (arrowheads). (Y) A graph showing the ratio of innervated HCs to total HCs in wildtype and *vangl2*^*m209*^ neuromasts. The number of neuromasts: WT, n = 11; *vangl2*^*m209*^, n = 13. N.S., student's t-test.

DOI: https://doi.org/10.7554/eLife.35796.021

The following source data is available for figure 8:

**Source data 1.** Ratio of Emx2-positive HCs within a neuromast.
DOI: https://doi.org/10.7554/eLife.35796.022
**Source data 2.** Ratio of innervated HCs to total number of HCs.
DOI: https://doi.org/10.7554/eLife.35796.023

neuromasts only respond to stimulation in the P > A direction because only P > A HCs are present. Likewise, neurons of *emx2* gof neuromasts only respond to hair bundle stimulation in the A > P direction.

Despite the increase in the amplitude of post-synaptic calcium signals within the entire afferent terminal in *emx2* mutant compared to controls, this increase did not translate into more evoked spikes at the level of a single afferent neuron, even though some of the neurons innervated more HCs than normal. What could account for these seemingly paradoxical results? It is possible that the increase in calcium signaling is not sufficient to elicit a change in spike number at the cell body level. It is also possible that due to the variability in our electrophysiology data, it may not be possible to resolve an overall increase in spike number in *emx2* mutant neurons. For example, the average number of spikes recorded from a single wildtype neuron shows a large range from 1 to 21 spikes (averages for 10 repeated 500 ms stimuli). It is not clear whether these differences represent a heterogenous population of neurons with different receptive fields. For example, if a single neuron innervates multiple neuromasts (large receptive field) it may only show a few spikes when only one of its neuromasts is stimulated, whereas a neuron that innervates only one neuromast (small receptive field) may respond with a large number of spikes when that neuromast is stimulated. Without the ability to further delineate the diverse types of neurons in wildtype and compare them directly to their counterparts in *emx2* mutants, it is not possible to determine whether the increase in post-synaptic calcium affected evoked spike numbers in the neuronal cell body. Furthermore, while our anatomical and physiological results indicate that afferent neurons in the *emx2* mutants are responding in a predicted manner, our results cannot distinguish functional differences between major versus minor afferents.

## Neuronal selectivity is independent of hair bundle orientation

The observed changes in neuronal innervation patterns in the *emx2* mutants are consistent with the changes in hair bundle orientation, suggesting that the transcriptional activity of Emx2 mediates both hair bundle orientation and selectivity of neuronal innervation. However, results from the *vangl2*^*m209*^ mutants suggest that neuronal selectivity is not dependent on hair bundle orientation per se. Although the mechanism of hair bundle misorientation in both *vangl2*^*m209*^ and *emx2* mutant mice may share a common pathway (*López-Schier and Hudspeth, 2006*; *Jiang et al., 2017*), the neuronal innervation pattern in *vangl2*^*m209*^ remains consistent with the Emx2 expression pattern of HCs. These results suggest that Emx2 regulates both hair bundle orientation and neuronal innervation patterns, whereas Vangl2 only regulates hair bundle orientation, suggesting that these two cellular processes are independently regulated.

Our results indicate that Emx2 expression in HCs of neuromasts is not affected in the *vangl2*^*m209*^ (*Figure 8*) and similarly, Vangl2 expression is not affected in *Emx2* ko mouse inner ears (*Jiang et al., 2017*), further supporting that these proteins are independently regulated. However, our genetic

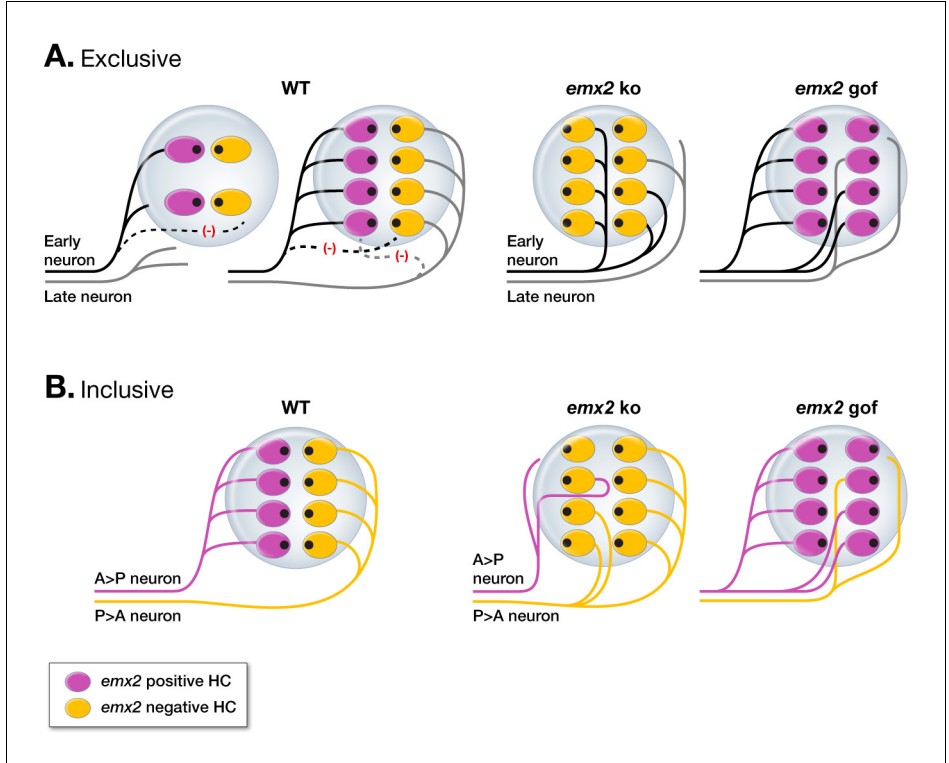

**Figure 9.** Models of the regulation of afferent innervation patterns in the zebrafish neuromast. (**A**) An Emx2 exclusive model. The first neuron (black) that reaches a normal neuromast innervates an Emx2-positive HC (magenta) due to specific downstream effectors of Emx2, which then set up the neuron's preference for similar HCs. As Emx2-positive HCs become occupied, a late arriving neuron (gray) starts to innervate the unoccupied Emx2-negative HCs (yellow). Subsequent cross-inhibition between the early and late neurons may help to maintain the segregated pattern of innervation. In *emx2* mutants, the first neuron (black) that reaches the neuromast becomes a major neuron. As HCs become occupied, a later arriving neuron (grey) finds fewer available HCs and becomes a minor neuron. (**B**) In the Emx2 inclusive model, two intrinsically different types of neurons innervate the neuromast. Due to their molecular determinants, an A > P neuron (magenta) prefers A > P HCs (magenta) and a P > A neuron (yellow) prefers P > A HCs (yellow), thus establishing the directional selectivity of neurons. In *emx2* ko neuromasts, a P > A neuron (yellow) expands its innervation pattern due to the increase in P > A HCs and becomes a major neuron, whereas an A > P neuron finds fewer P > A HCs and forms a few synapses with P > A HCs (or not). By contrast, in *emx2* gof neuromasts, an A > P neuron becomes a major fiber and a P > A neuron forms a minor neuronal fiber.

DOI: https://doi.org/10.7554/eLife.35796.024

analyses indicate that *vangl2* function is epistatic to *emx2* with regards to hair bundle orientation. These results are consistent with results in the mouse utricle indicating that a core planar cell polarity protein such as Vangl2, functions in an intercellular manner and regulates an intracellular signaling mechanism such as Emx2 (*Montcouquiol et al., 2006*; *Yin et al., 2012*; *Copley et al., 2013*; *Jiang et al., 2017*).

Taken together our results indicate that the restricted expression of Emx2 to half of the HCs within a neuromast reverses the hair bundle orientation by 180 degrees and allows the detection of water flow from the opposite direction (*Jiang et al., 2017*). In addition, the expression of Emx2 guides neuronal patterning and preserves the directional selectivity of the HCs. Thus, Emx2 regulates the two key components that confer directional selectivity in the peripheral lateral line system. Given the conserved role of Emx2 in mediating hair bundle orientation among mechanosensory organs of fish and mammals (*Jiang et al., 2017*), its role in regulating post-synaptic partner described here may apply to other neural systems where Emx2 is normally expressed such as the cortex and olfactory epithelium (*Cecchi and Boncinelli, 2000*; *Cecchi, 2002*; *Nédélec et al., 2004*).

# Materials and methods

## Key resources table

| Strains, reagent type or resource | Designation | Source or reference | Identifiers |
|---|---|---|---|
| Strains | Tg(myo6b:actb1-GFP)$^{vo8Tg}$ | Kindt et al. (2012) | RRID:ZFIN_ZDB-GENO-120926-20 |
| Strains | HGn39D | Nagayoshi et al. (2008) | RRID:ZFIN_ZDB-GENO-100113-50 |
| Strains | Tg(myo6b:emx2-p2a-nls-mCherry)$^{idc4Tg}$ | Jiang et al., 2017 | RRID:ZFIN_ZDB-GENO-170619-3 |
| Strains | emx2$^{idc5}$ | Jiang et al., 2017 | RRID:ZFIN_ZDB-ALT-170606-5 |
| Strains | vangl$^{m209}$ | Jessen et al. (2002) | RRID:ZFIN_ZDB-GENO-100615-1 |
| Strains | Tg(hsp70l:GCaMP6s-CAAX-SiLL1)$^{idc8Tg}$ | Zhang et al. (2018) | |
| Chemical compound | ethyl 3-aminobenzoate methanesulfonate | Sigma | MS-222 |
| Chemical compound | Alexa Fluor 647 phalloidin | Thermo Fisher Scientific | A22287 |
| Chemical compound | FM 4–64 | Thermo Fisher Scientific | T13320 |
| Protein | α-bungarotoxin | Tocris | 2133 |
| Antibody | rabbit anti-Emx2 | Trans Genic | KO609 |
| Antibody | mouse anti-RibeyeB | Sheets et al. (2011) | |
| Antibody | mouse anti-Maguk clone K28/86 | NeuroMab | |

## Zebrafish strains

Adult zebrafish (Danio rerio) were maintained with a 14 hr light, 10 hr dark cycle. Zebrafish protocols were approved by the Animal User Committee at the National Institutes of Health (Animal study protocol #1362–13). Zebrafish larvae were raised in E3 embryo media as follows: 5 NaCl, 0.17 KCl, 0.33 CaCl2, and 0.33 MgSO4, buffered in HEPES, at 28.5°C. Previously described transgenic zebrafish strains used in this study include the following: Tg(myo6b:actb1-GFP)$^{vo8Tg}$ (RRID:ZFIN_ZDB-GENO-120926-20), Tg(cntnap2a)$^{nkhgn39dEt}$ or HGn39D (RRID:ZFIN_ZDB-GENO-100113-50), Tg(myo6b:emx2-p2a-nls-mCherry)$^{idc4Tg}$ (RRID:ZFIN_ZDB-GENO-170619-3), emx2$^{idc5}$ (emx2 ko) (RRID:ZFIN_ZDB-ALT-170606-5), vangl$^{m209}$ (RRID:ZFIN_ZDB-GENO-100615-1), Tg(hsp70l:GCaMP6s-CAAX-SiL-L1)$^{idc8Tg}$ (Solnica-Krezel et al., 1996; Jessen et al., 2002; Nagayoshi et al., 2008; Kindt et al., 2012; Jiang et al., 2017; Zhang et al., 2018). F1 to F3 generation of emx2 gof (myo6b:emx2-p2a-nls-mCherry) were used in our study, and incomplete penetrance of the hair bundle phenotype in neuromasts from F2 and F3 generations have been observed. Larvae were used between 3–6 dpf.

## DNA injection and screening of transgenic fish

To label a single afferent neuron, neuroD:tdTomato plasmid was used, in which a 5 kb minimal promoter of neurod drives tdTomato expression (Toro et al., 2015). NeuroD:tdTomato plasmid at a concentration of 50 ng/ul was pressure injected to one-cell stage zebrafish embryos. Larvae were screened at 2–3 dpf (days post fertilization) for tdTomato expression in the posterior lateral line axon using a SteREO Discovery V20 microscope with an X-Cite 120 external fluorescent light source (EXFO Photonic Solutions Inc., Ontario, Canada). After selection of candidates, definitive expression in the posterior lateral line axon was ascertained using a 63X oil immersion objective on a Zeiss LSM780 confocal microscope (Carl Zeiss Ltd. Cambridge, UK).

## Live imaging

Zebrafish larvae as 3–4 dpf were anesthetized with 0.03% ethyl 3-aminobenzoate methanesulfonate (Sigma, #MS-222, St Louis, MO) and embedded in 1.5% low melting point agarose containing 0.01% tricaine, and then imaged at 63X using a Zeiss LSM780 confocal microscope.

## Immunofluorescence and phalloidin staining

For staining zebrafish larvae, 3–6 dpf embryos were fixed with 4% paraformaldehyde in PBS for 3.5 hr at 4°C. Post-fixed larvae were rinsed with 0.05% Tween-20 PBS (PBT) and treated with pre-chilled acetone at - 20°C. Then, larvae were incubated with a blocking solution (2% goat serum, 1% BSA in PBT) for 2 hr at room temperature or overnight at 4°C, followed by incubation with primary antibodies diluted in blocking solution for 2 hr at room temperature or overnight at 4°C. Larvae were washed four times with PBT for 5 min each before incubating with secondary antibodies at 1:1000 dilution in blocking solution for 2 hr at room temperature. Then, larvae were washed and mounted with Anti-fade and imaged with a Zeiss LSM780 confocal microscope. For whole-mount immunofluorescence labeling, we used the following primary antibodies: rabbit anti-Emx2 (1:250; KO609, Trans Genic, Fukuoka, Japan), mouse anti-RibeyeB (1:1000) (*Sheets et al., 2011*), and mouse anti-Maguk clone K28/86 (1:500; NeuroMab, Davis, CA, RRID:AB_2292909). To detect F-actin, we stained with fluorescein or Alexa Fluor 647 phalloidin (1:50; Thermo Fisher Scientific, Carlsbad, CA, RRID:AB_2620155).

## Zebrafish immobilization and hair cell stimulation

To suppress muscle activity and movement for functional analyses, larvae were anesthetized with 0.03% ethyl 3-aminobenzoate methanesulfonate (MS-222) in E3, and mounted onto Sylgard (Dow Corning) filled dishes using tungsten pins. After larvae were anesthetized, they were then microinjected in the heart with 125 µM α-bungarotoxin (Tocris) to suppress muscle activity. After microinjection, larvae were then rinsed thoroughly to remove MS-222, and maintained in extracellular solution in mM: 130 NaCl, 2 KCl, 2 CaCl$_2$, 1 MgCl$_2$ and 10 HEPES, pH 7.3, 290 mOsm. To stimulate lateral-line HCs we used a pressure clamp (HSPC-1, ALA Scientific, New York) attached to a glass or water-jet micropipette (inner tip diameter ~30–50 µm) filled with extracellular solution. The waterjet was positioned (MP-265, Sutter Instruments) approximately 100 µm from a given neuromast. The waterjet was placed in focus in the plane of the kinocilial tips. Displacement (~5 µm) of the kinocilial tips was verified by measuring the distance moved during the displacement. To trigger the waterjet during recordings of lateral-line afferents (see below), the pressure clamp was driven by a voltage command delivered by the recording amplifier. For afferent calcium imaging experiments, the pressure clamp was driven by a voltage step command. This voltage command was generated in the Prairie View (Bruker Corporation) imaging software during calcium imaging.

## Electrophysiological recordings from lateral-line afferents

The method for recording postsynaptic action currents has been described previously (*Olt et al., 2016*; *Sheets et al., 2017*). All recordings were performed in extracellular solution (see above), on afferent neurons innervating zebrafish primary neuromasts (L1–L4). For the extracellular recordings, borosilicate glass pipettes were pulled (P-97, Sutter Instruments) with a long taper to obtain resistances between 5 and 10 MΩ in extracellular solution. Signals were collected with a Multiclamp 700B, a Digidata 1550 data acquisition board, with pClamp10 software (Molecular Devices, LLC). Extracellular currents were acquired from lateral-line afferent neuron in the loose-patch configuration (seal resistances ranged from 20 to 80 MΩ in extracellular solution). Recordings were done in voltage-clamp mode, sampled at 50 µs/pt, and filtered at 1 kHz. To identify the neuromast innervating an afferent neuron, the fluid-jet was used to progressively stimulate primary neuromasts of the posterior lateral line until phase-locked spiking was detected.

## Afferent calcium imaging

Calcium imaging experiments were performed in extracellular solution (see above). Afferent calcium measurements were made at postsynaptic sites directly beneath HCs using *hsp70l:GCaMP6s-CAAX-SiLL1* transgenic fish. Acquisitions were made on an upright Swept-field confocal system (Bruker Corporation) with a 60X 1.0 NA CFI water-immersion objective (Nikon, Tokyo, Japan). The Swept-field microscope used a Rolera EM-C2 EMCCD camera (QImaging) controlled using Prairie view (Bruker Corporation). GCaMP6s-CAAX was excited using a 488 nm solid-state laser. To image calcium activity within the entire post-synaptic process simultaneously across the Z-axis, we used piezoelectric motor (PICMA P-882.11–888.11 series, PI instruments) attached to the objective. The piezo allowed rapid imaging in five planes along the Z-axis every 2 µm at a 50 Hz frame rate, enabling a 10 Hz

volume rate. Our five plane Z-stacks were later projected into one plane for further image processing and quantification (see below). After functional calcium imaging, each larva was incubated in 3 µM FM 4–64 (Thermo Fisher Scientific) in E3 for 30 s, followed by 3 E3 washes, to label HCs. Confocal images of afferent process and FM 4–64 labeled HCs were then acquired on an upright Nikon C2 laser-scanning confocal microscope using a 60X 1.0 NA water objective lens. Appropriate solid-state lasers were used to image and excite GCaMP6s and FM 4–64.

## Functional signal analysis

Afferent electrophysiology recordings were analyzed with software written in Igor Pro (Wavemetrics) and were plotted with Igor Pro and Prism 7 (Graphpad). For afferent GCaMP6s-CAAX calcium measurements, the raw images were processed using a custom program in Matlab (MathWorks, *Source code 1–3*). For this analysis, we removed the first 10 images to reduce the effect of photobleaching. Then the raw images were registered to reduce movement artifacts by applying efficient subpixel image registration based on cross-correlation (*Zhang et al., 2016*). Then the spatial calcium signals were converted to heat maps as previously described (*Zhang et al., 2016*). Briefly, we first computed the baseline image (or reference image or $F_0$) by averaging the images during the pre-stimulus period. Then the baseline image ($F_0$) was subtracted from each image acquired to generate images that represent the relative change in fluorescent signal, $\Delta F$-$F_0$ or $\Delta F$. To visualize the fluorescence intensity changes during stimulation, $\Delta F$ signals during the stimulus period were averaged, scaled and encoded by color maps with red indicating an increase in signal intensity. The color maps were then superimposed onto the $F_0$ baseline grayscale image, in order to visualize the spatial fluorescence intensity changes in the afferent processes during stimulation.

To quantify changes in afferent calcium signals we analyzed the registered images in ImageJ. To quantify the signals in an entire afferent process innervating a neuromast, we applied an inclusive threshold to capture all the pixels within the process. The same threshold was applied to all wildtype and *emx2* mutant processes. Each threshold was used to generate a ROI to compute mean intensity changes ($\Delta F/F_0$) within the entire process throughout the acquisition. To quantify afferent calcium signals per HC, circular ROIs with a diameter of ~3 µm (12 pixels with 268 nm per pixel) were placed on the afferent process beneath each HC within a neuromast. After creating these ROIs, we then computed and plotted the mean intensity changes ($\Delta F/F_0$) within each ROI during the acquisition period. Quantification of $\Delta F/F_0$ magnitudes was performed using scripts written in Matlab (The Mathworks) as described previously (*Hilliard et al., 2005*). The signal magnitude was defined as the peak value of intensity change upon stimulation.

## Image processing

To quantify the colocalization between RibeyeB and nerve endings, the number of RibeyeB puncta that were contacted by a single-labeled neuron per HC were counted. To further quantify this, a Mander's colocalization analysis between RibeyeB and tdTomato-positive nerve endings was conducted by first placing 6.5 µm² ROIs beneath each HC innervated by a single labeled-afferent neuron and the area of overlap was measured. These measurements were made on selected 1µm optical sections centered on the ribbons to avoid overlap with the tdTomato-positive nerve fiber in different planes. Then, the Mander's coefficient was calculated using ImageJ. These values were averaged to determine the amount of overlap per neuromast. To quantify the number of RibeyeB and Maguk puncta, we used ImageJ to z-project all planes containing synapses. Then, we used ImageJ to apply the same background subtraction and threshold to each immunostain. After this processing we counted the total number of puncta, as well as complete and incomplete synapses by eye. For these counts, images were scored blinded. For RibeyeB and Maguk colocalization, we calculated the Mander's coefficient to determine the amount of Maguk label that colocalized with Ribeye label, due to the high background in Maguk immunostain.

For our figures, images were processed and adjusted for brightness and contrast with ImageJ. Graphs were generated by Prism. Figures were assembled with Adobe Photoshop CS5.

## Statistical analyses

To compare the afferent calcium signals per HC, a Kruskal-Wallis test in Prism was used. To compare the average afferent calcium signal per afferent process, number of spikes a one-way ANOVA in

Prism was used. To compare complete and incomplete synapses, colocalization coefficients and number of innervated HCs between wildtype, emx2 ko and emx2 gof, a one-way ANOVA in excel was used. To compare the number of innervated HCs or Emx2-positive HCs between WT and $vangl2^{m209}$ mutants, a student's t-test was used. In all graphs, errors bars represent standard error of the mean (SEM).

## Acknowledgements

We would like to thank Alisha Beirl for technical assistance, Jennifer Wang for quantification of RibeyeB and Maguk staining, Dr. Lopez-Schier for the HGn39D strain and ZIRC for the $vangl2^{m209}$ strain. We are also grateful to Drs. Lisa Cunningham and Thomas Friedman and members of our laboratories for critical reading of the manuscript. This study is supported by NIDCD Intramural program awards to DKW and KK.

## Additional information

### Funding

| Funder | Grant reference number | Author |
|---|---|---|
| National Institute on Deafness and Other Communication Disorders | Intramural Research Program Grant 1ZIADC000085-01 | Katie Kindt |
| National Institute on Deafness and Other Communication Disorders | Intramural Research Program Grant 1ZIADC000021 | Doris K Wu |

The funders had no role in study design, data collection and interpretation, or the decision to submit the work for publication.

### Author contributions

Young Rae Ji, Conceptualization, Resources, Data curation, Formal analysis, Validation, Investigation, Writing—original draft, Writing—review and editing; Sunita Warrier, Resources, Data curation, Investigation; Tao Jiang, Data curation, Methodology; Doris K Wu, Resources, Supervision, Funding acquisition, Validation, Writing—original draft, Project administration, Writing—review and editing; Katie S Kindt, Data curation, Software, Funding acquisition, Methodology, Resources, Supervision, Validation, Project administration, Writing – original draft, review and editing

### Author ORCIDs

Young Rae Ji http://orcid.org/0000-0002-8825-9783
Doris K Wu http://orcid.org/0000-0002-1400-3558
Katie S Kindt https://orcid.org/0000-0002-1065-8215

### Decision letter and Author response

Decision letter https://doi.org/10.7554/eLife.35796.031
Author response https://doi.org/10.7554/eLife.35796.032

## Additional files

### Supplementary files

• Source code 1. Source code of Matlab software 1.
DOI: https://doi.org/10.7554/eLife.35796.025

• Source code 2. Source code of Matlab software 2.
DOI: https://doi.org/10.7554/eLife.35796.026

• Source code 3. The note of Matlab software.
DOI: https://doi.org/10.7554/eLife.35796.027

• Transparent reporting form

DOI: https://doi.org/10.7554/eLife.35796.028

## Data availability

All data generated or analysed during this study are included in the manuscript. Source data files have been provided for Table 1 and Figure 2, 4, 5, 6, 7, and 8.

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
