## [Decision Letter]

Thank you for submitting your article "Directional selectivity of afferent neurons in zebrafish neuromasts is regulated by Emx2 in presynaptic hair cells" for consideration by *eLife*. Your article has been reviewed by three peer reviewers, and the evaluation has been overseen a Senior/Reviewing Editor. The reviewers have opted to remain anonymous. The reviewers all think very highly of the manuscript and request relatively minor revisions, detailed below.

Summary:

In this study, Ji et al. present evidence that innervation of zebrafish lateral line hair cells is controlled by Emx2. Through a combination of gain and loss of function approaches, imaging and in vivo physiology, the authors demonstrate that manipulating Emx2 expression alters innervation pattern dependent on directional polarity. Analysis of altered Emx2 in conjunction with the planar cell polarity regulator Vangl2 demonstrate that innervation can be uncoupled from the regulation of bundle polarity. This is an interesting and significant study that provides new insights into mechanisms of circuit assembly and highlights a novel and likely conserved role of the transcription factor Emx2 in generating target identity and/or target-derived signals during hair cell synapse formation.

Minor points:

1) Figure 3.

(L1-L3) Please indicate how many hair cells were contacted by minor afferents, as was done for (H1-H3).

(T1-T2) Please indicate how many hair cells were contacted by minor afferents, as was done for (P1-P3).

2) Subsection “Both major and minor afferent nerve endings are associated with presynaptic specializations in *emx2* mutants” and Figure 4. It is not clear what is meant by "incomplete synapses" or whether non-overlapping punctae were increased in *emx2* mutants. Quantification of the overlap between RibeyeB and tdTomato-labeled afferent fiber would be informative.

3) Figure 5 and related text. To help separately evaluate of pre- and post- synaptic specializations, it would be informative to calculate the fractional overlap between RibeyeB and Maguk using Manders' Colocalization Coefficients (Dunn et al., 2011). Moreover, were total numbers of ribeye and Maguk punctae per neuromast changed in *emx2* lof and gof mutants?

4) Figure 8Q-X. For better clarity, the five Emx2+ve hair cells should be circled *and* numbered (T, U), and the corresponding afferent contact should be indicated by arrowheads *and* the matching number (V, W, X).

5) In some figures (Figures 1, 4, 5, 8), white arrows are used both to indicate hair bundle orientation, and to point at a specific hair cell, which was a bit confusing. Suggest using a different color to denote hair cells.

6) Subsection “Both major and minor afferent nerve endings are associated with presynaptic specializations in *emx2* mutants”, the Sheets et al. reference was inserted at an odd place.

7) Subsection “Exclusive or inclusive role of Emx2 in neuronal selectivity”, end of first paragraph. The sentence was unclear. Was the total number of neurons innervation one neuromast decreased in the *emx2* mutant?

8) The authors provide detailed information about tests and sample number. While it will be unlikely to change conclusions, the authors should correct for multiple testing in their statistical analysis. Instead of using a t-test with categories greater than two, they should perform ANOVA with posthoc testing.

---

## [Author Response]

Minor points:1) Figure 3.(L1-L3) Please indicate how many hair cells were contacted by minor afferents, as was done for (H1-H3).(T1-T2) Please indicate how many hair cells were contacted by minor afferents, as was done for (P1-P3).

The number of hair cells contacted by minor afferents were described in the original Results and Legend sections. These cells are now marked by white arrowheads in L1-L3 and T1-T2 in the revised figure.

2) Subsection “Both major and minor afferent nerve endings are associated with presynaptic specializations in emx2 mutants” and Figure 4. It is not clear what is meant by "incomplete synapses" or whether non-overlapping punctae were increased in emx2 mutants. Quantification of the overlap between RibeyeB and tdTomato-labeled afferent fiber would be informative.

The term “incomplete synapses” has been defined in the last sentence of the subsection “Both major and minor afferent nerve endings are associated with presynaptic specializations in *emx2* mutants”.

Quantification of overlap between RibeyeB and tdTomato-labeled afferents were conducted and Manders’ coefficients were calculated. We observed a significant decrease in the number and Manders’ coefficients for RibeyeB and nerve ending overlap in minor fibers of emx2 mutants. The results are described in the Results section (subsection “Both major and minor afferent nerve endings are associated with presynaptic specializations in *emx2* mutants”) and the data are shown in Figure 4—source data 1 and 2.

3) Figure 5 and related text. To help separately evaluate of pre- and post- synaptic specializations, it would be informative to calculate the fractional overlap between RibeyeB and Maguk using Manders' Colocalization Coefficients (Dunn et al., 2011). Moreover, were total numbers of ribeye and Maguk punctae per neuromast changed in emx2 lof and gof mutants?

We conducted the fractional overlap data between RibeyeB and Maguk shown in Figure 5M using Manders’ colocalization coefficients. No statistical significance differences were detected between mutants and wildtype. The data are now shown as Figure 5—source data 1 and 2.

The total number of RibeyeB and Maguk puncta per neuromast are increased in *emx2* mutants compared to wildtype due to an increase in the number of solitary RibeyeB and Maguk puncta (Figure 5N). The raw data are now shown in Figure 5—source data 3 and 4.

4) Figure 8Q-X. For better clarity, the five Emx2+ve hair cells should be circled and numbered (T, U), and the corresponding afferent contact should be indicated by arrowheads and the matching number (V, W, X).

The figure has been revised accordingly.

5) In some figures (Figure 1, 4, 5, 8), white arrows are used both to indicate hair bundle orientation, and to point at a specific hair cell, which was a bit confusing. Suggest using a different color to denote hair cells.

The color for arrows indicating hair bundle orientation has been changed to cyan color.

6) Subsection “Both major and minor afferent nerve endings are associated with presynaptic specializations in emx2 mutants”, the Sheets et al. reference was inserted at an odd place.

The citation is now moved (see end of subsection “Both major and minor afferent nerve endings are associated with presynaptic specializations in *emx2* mutants”).

7) Subsection “Exclusive or inclusive role of Emx2 in neuronal selectivity”, end of first paragraph. The sentence was unclear. Was the total number of neurons innervation one neuromast decreased in the emx2 mutant?

The sentence referred to the smaller number of single neurons with a minor innervation pattern. We have clarified this point in the subsection “Exclusive or inclusive role of Emx2 in neuronal selectivity”.

Since our experimental design focused on the innervation pattern of a single-labeled afferent neuron within a neuromast, we cannot directly address the total number of neurons per neuromast in either wildtype or *emx2* mutants. Similar single-neuron labeling results suggest that there are four neurons per neuromast, two neurons for hair cells of each orientation (Pujol-Marti et al., 2014). However, results of retrograde labeling using fluorescent dyes suggest that the total number of afferent neurons can be quite variable among neuromasts, between 4 to 6 neurons (Haehnel et al., 2012). Therefore, the total number of afferent neurons per neuromast in both wildtype and *emx2* mutants requires further experimental investigation.

8) The authors provide detailed information about tests and sample number. While it will be unlikely to change conclusions, the authors should correct for multiple testing in their statistical analysis. Instead of using a t-test with categories greater than two, they should perform ANOVA with posthoc testing.

ANOVA with posthoc testing was conducted as suggested.